# Ten-years cardiovascular risk among Bangladeshi population using non-laboratory-based risk chart of the World Health Organization: Findings from a nationally representative survey

**Abu Abdullah Mohammad Hanif**[1], **Mehedi Hasan**[1], **Md Showkat Ali Khan**[1], **Md. Mokbul Hossain**[1], **Abu Ahmed Shamim**[1], **Moyazzam Hossaine**[1], **Mohammad Aman Ullah**[2], **Samir Kanti Sarker**[2], **S. M Mustafizur Rahman**[2], **Md Mofijul Islam Bulbul**[2], **Dipak Kumar Mitra**[3], **Malay Kanti Mridha**[1] *

1 Centre for Non-Communicable Diseases and Nutrition, BRAC James P Grant School of Public Health, BRAC University, Dhaka, Bangladesh, 2 National Nutrition Services (NNS), Institute of Public Health Nutrition (IPHN), Dhaka, Bangladesh, 3 North-South University, Dhaka, Bangladesh

* malay.mridha@bracu.ac.bd

## Abstract

The World Health Organization (WHO) has recently developed a non-laboratory based cardio-vascular disease (CVD) risk chart considering the parameters age, sex, current smoking status, systolic blood pressure, and body mass index. Using the chart, we estimated the 10-years CVD risk among the Bangladeshi population aged 40–74 years. We analyzed data from a nationally representative survey conducted in 2018–19. The survey enrolled participants from 82 clusters (57 rural, 15 non-slum urban, and 10 slums) selected by multistage cluster sampling. Using the non-laboratory-based CVD risk chart of the World Health Organization (WHO), we categorized the participants into 5 risk groups: very low (<5%), low (5% to <10%), moderate (10% to <20%), high (20% to <30%) and very high (> = 30%) risk. We performed descriptive analyses to report the distribution of CVD risk and carried out univariable and multivariable logistic regression to identify factors associated with elevated CVD risk (> = 10% CVD risk). Of the 7,381 participants, 46.0% were female. The median age (IQR) was 59.0 (48.0–64.7) years. Overall, the prevalence of very low, low, moderate, high, and very high CVD risk was 34.7%, 37.8%, 25.9%, 1.6%, and 0.1%, respectively. Elevated CVD risk (> = 10%) was associated with poor education, currently unmarried, insufficient physical inactivity, smokeless tobacco use, and self-reported diabetes in both sexes, higher household income, and higher sedentary time among males, and slum-dwelling and non-Muslim religions among females. One in every four Bangladeshi adults had elevated levels of CVD risk, and males are at higher risk of occurring CVD events. Non-laboratory-based risk prediction charts can be effectively used in low resource settings. The government of Bangladesh and other developing countries should train the primary health care workers on the use of WHO non-laboratory-based CVD risk charts, especially in settings where laboratory tests are not available.

**Data Availability Statement:** All relevant data are within the paper and its Supporting Information files.

**Funding:** The study was funded by the National Nutrition Services (NNS), Institute of Public Health Nutrition, Ministry of Health and Family Welfare, Government of Bangladesh. Besides, salaries and administrative support for some of authors came from the National Institute for Health Research (NIHR) (16/136/68) using UK aid from the UK Government to support global health research, and by Wellcome Trust (212945/Z/18/Z). The views expressed in this publication are those of the author(s) and not necessarily those of the NIHR or the UK Department of Health and Social Care.

**Competing interests:** Some of the representatives of the Ministry of Health and Family Welfare, who approved the funding of the study were involved with the Technical Advisory Group. Though they had opinions about some maternal and child health and nutrition indicators, they did not have any role in the design, conduct, data analysis, and manuscript writing of the adolescent component of the study. However, this does not alter our adherence to PLOS ONE policies on sharing data and materials.

## Introduction

The world has been experiencing a massive burden of noncommunicable diseases (NCD) for the last few decades, and the trend is steadily upward. Noncommunicable diseases (NCDs) claim a total of 41 million lives every year, which is equivalent to 71% of all global deaths [1]. About 37% of these deaths occur between the ages of 30 and 69 years, and 85% of these premature deaths occur in low and middle-income countries [1]. Among the NCDs, cardiovascular diseases (CVD) such as coronary heart diseases and stroke are most common and were responsible for an estimated 17·8 million deaths in 2017, and 75% of these deaths were in the low-income and middle-income countries [2]. In 2040, eight of the top ten causes of death worldwide will be NCDs with coronary heart diseases, and stroke will continue to be in the first and the second places, respectively [3]. South Asia has the highest total burden of CVD partly due to its massive population size and early onset of CVD in this population [4]. Between 1990 and 2010, there was a 73% increase in the years of life lost in the South Asian region, compared to a 30% global increase in the same period due to CVDs [5]. In Bangladesh, NCDs are responsible for 67% of all deaths; and an estimated 30% of the total deaths are caused by CVDs [6].

CVD is an umbrella term coined for conditions affecting the heart or blood vessels. CVD causes arterial damage in the major organs such as the brain, heart, kidney, and eyes [7]. However, most CVDs are preventable (through addressing the) modifiable risk factors; through reducing or preventing behavioral risk factors such as smoking and smokeless tobacco use, unhealthy diet, insufficient physical activity, overweight, and obesity [8]. One way to reduce the burden of CVDs is to identify people with the risk of developing CVDs due to hypertension, diabetes, dyslipidemia, and implementing appropriate management protocols [1]. Assessment of CVD risk based on multiple risk factors rather than individual risk factors brings more accuracy and allows cost-effective management and treatment of CVD [9].

In 2019, the World Health Organization (WHO) revised the 2007 CVD risk chart with the help of a risk-chart working committee using data from 21 global regions [10]. The committee revised the two previous laboratory-based charts and proposed a new non-laboratory-based chart. The WHO Laboratory-based CVD risk assessment chart predicts the 10-years risk of fatal or non-fatal CVD events using data on age, sex, smoking status, systolic blood pressure, and total serum cholesterol, along with the history of diabetes. The WHO non-laboratory-based CVD risk chart was designed for resource-poor settings. This chart predicts ten-years CVD risk using the information on age, sex, smoking status, systolic blood pressure, and body mass index. Bangladesh has recently incorporated these revised CVD risk charts in its 'NCD management protocol' and has started training the primary health care workers [11]. However, as the laboratory-based charts need data on blood sugar and total serum cholesterol, the use of the laboratory-based charts can be impossible in settings where these tests are unavailable. Therefore, in resource-poor settings, the non-laboratory-based chart can help the health care workers in screening, primary management, counseling, and referral of the patients at risk of CVD. However, there is a lack of nationally representative data on the prevalence of ten-years CVD risk in Bangladesh. In this study, we aimed to assess the prevalence of ten-years CVD risk among the 40–74 years old population of Bangladesh using the non-laboratory-based CVD risk assessment of the WHO. Our study will help the government to plan specific NCD interventions targeted to people at different risk levels of CVD.

## Materials and methods

### Study design and site

The government of Bangladesh has been implementing the Food Security and Nutrition Surveillance Project (FSNSP) among women and children since 1990 [12]. In the 2018–19 round

of this surveillance, we included other population groups, e.g., adolescent boys and girls, adult males, and elderly people. The FSNSP is aimed to generate nationally and divisionally representative estimates of various nutritional and health-related variables. The 2018–2019 round was conducted between October 2018 and October 2019. In this round, we enrolled participants from rural, non-slum urban, and slum areas in all eight administrative divisions of Bangladesh. We enrolled in study participants from 82 randomly selected clusters (57 rural, 15 non-slums urban, and 10 slums) from all over Bangladesh. In the main survey, we enrolled participants from six age groups: children (0 to <5 years), adolescent girls (10–19 years), adolescent boys (10–19 years), adult women (20–59 years), adult men (20–59 years) and elderly people (60 years and above).

## Sample size and sampling techniques

We used a multistage cluster sampling to determine the sample size for the selected indicators with prevalence (p) ranged from 4% to 98% to generate national and divisional estimates. Considering the type I error, $\alpha = 0.05$; allowable margin of error, $d = 0.05$ (or $d = p/2$ if $p \leq 0.1$); design effect, DEF = 1.61, we calculated a sample size of 62 individuals from each cluster for each age group.

We selected the study sites in rural, urban, and slum areas applying different sampling techniques. For rural areas, we randomly selected two districts from each of the divisions as the first stage of the multistage sampling, and from each of the selected districts, we chose one sub-districts. We then selected two unions (smallest administrative units of Bangladesh) from each of the selected sub-districts. After consulting with the local government officials, the field coordinator identified and mapped the villages/mouzas/geographically demarcated segments with 250–400 households in the selected unions. Finally, we randomly chose two of the listed village/mouza/segments from each union as the study clusters.

In the non-slum urban areas, we used the population proportion of the Bangladesh Bureau of Statistics (BBS) 2011 census to select the required study clusters [13]. We randomly selected 15 wards (1–2 wards/division) from the city corporations. The field coordinator identified and mapped the Mahalla (similar to the villages) with more than 250 households. They further subdivided the mahallas with >500 households into smaller geographically defined segments of ~250 households. We randomly selected one segment from each of the selected wards as the non-slum urban clusters.

In the slum areas, we took guidance from the Census of Slum Areas and Floating Population 2014 to select the study clusters [14]. The field coordinator identified and mapped the slums with ≥300 households and further sub-divided the slums with >500 households into smaller segments. We then randomly selected two segments or slums from the Dhaka and Chattogram division and one from each of the other six divisions as the slum study clusters.

For each cluster, research assistants first listed all households and their members according to the age groups. If any household had more than one person of a specific age group, we randomly selected one member of that age group from that household and thus came up with a sampling frame for that cluster. A statistician then selected 80 individuals from the list using Simple Random Sampling to enroll 62 participants from each age group from a cluster.

## Data collection and measurements

Five data collection teams—each comprised of 4–5 Research Assistants (RA) and one Project Officer (PO) were deployed for data collection. We used a structured questionnaire, developed initially in English and later translated into Bengali, to collect the data using face-to-face interviews. RAs directly entered all the data from face-to-face interviews and physical

measurements into the tablet computers (Samsung Galaxy Tab A7) using a customized SurveyCTO application [Dobility, Inc.]. They uploaded all the collected data to the server at the end of the day. We measured height (using a locally made portable stadiometer), weight (using TANITA UM-070 weighing scale), waist circumference (using measuring tape), and blood pressure (using Omron HEM 7120) of the study participants. As specified in the Food and Nutrition Technical Assistance (FANTA) anthropometry manual, the WHO guideline was followed to take anthropometric measurements [15]. We measured the weight to the nearest 0.1 kg and the height to the nearest 0.1 cm. Waist circumference was measured to the nearest 0.1 cm. RAs took two measurements of weight, height, and waist circumference. If the difference between the first two measurements were >0.1 kg for weight and >0.5 cm for height and waist circumference, they took the third measurement. RAs ensured that the study participant was resting for at least 15 minutes before taking blood pressure, and a three minutes interval was programmed between two subsequent blood pressure readings. We instructed the research assistants to take a third measurement only if the difference between the first two measurements was ≥10 mmHg for systolic and/or diastolic blood pressure.

## The WHO cardiovascular diseases risk charts

We used the WHO non-laboratory-based CVD risk chart developed for 5 south Asian countries (Bangladesh, Bhutan, Nepal, India, and Pakistan) to estimate the risk of fatal or non-fatal cardiovascular events, such as myocardial infarction and stroke in ten years period [10]. In 2019, the WHO CVD risk charts working group constructed this chart for the resource-poor settings and revised the previous laboratory-based charts [10]. The parameters used in the risk algorithm of the non-laboratory-based chart were age (in years), sex (male vs. female), smoking status (no vs. yes), body mass index (BMI) as weight in kg divided by height in squared-meter, and systolic blood pressure (SBP) in mmHg. Although this chart can provide the exact risk score of individuals aged 40 to 74 years, it can also stratify the people of this age group into six groups based on the calculated risk score–very low risk (<5%), low risk (5% to <10%), moderate risk (10% to <20%), high risk (20% to <30%) and very high risk (≥30%) to implement different management protocols [16].

## Explanatory variables

We defined the outcome variable as elevated CVD risk if the total CVD risk score was ≥10% and lower CVD risk if the risk score was <10%. We listed the potential factors associated with elevated CVD risk (CVD risk ≥10%) among the population of Bangladesh based on literature review and considering data availability from this survey. The sociodemographic variables included the place of residence (rural, non-slum urban, and slum), education (no formal education, up to 5 years, up to 10 years and >10 years), household income, marital status (currently married vs. never married or divorced or widowed or separated), and religion (Muslim vs. non-Muslims). Among the behavioral variables were physical activity, sedentary time (0 to 240 minutes/241 to 360 minutes/>360 minutes), fruits and vegetable consumption (≥5 servings/day vs. <5 servings), and smokeless tobacco consumption status (no vs. yes). We considered a person physically active (during work, transport, and recreational activities) if he or she reported at least 150 minutes of moderate-intensity physical activity per week or 75 minutes of vigorous physical activity per week or equivalent [17]. Self-reported diabetes and waist circumference were the clinical and anthropometric and factors. Central obesity was defined as the waist circumference of ≥90 cm in males and ≥80 cm in females [18]. For this analysis, we considered the mean of the two closest measurements of all anthropometric variables and blood

pressure. Self-reported diabetes was recorded if any respondent reported that a trained health care provider ever told him or her that he or she had diabetes.

## Quality assurance and control

We provided extensive training to the RAs and the POs on data collection, physical measurements, as well as calibration and maintenance of data collection instruments. We also organized standardization sessions to check the readiness of the RAs for data collection activities. We field-tested the questionnaire, modified it, and refreshed the RA and POs based on the feedback from the field testing. The POs directly observed 5% of the interviews and re-interviewed another 5% of the randomly selected study participants within 48 hours of the initial interview to ensure data quality. We also performed interim analyses to check the data quality. The RAs routinely calibrated the data collection instruments.

## Statistical analysis

We performed all the data management, cleaning, and analysis using Stata 15.1 (Stata Corp, College Station, TX, USA) [19]. All the characteristics of the study participants in this analysis were reported as categorical variables. As the males and females differed by the distribution of cardiovascular disease risk, we carried out univariable and multivariable logistic regression analysis to identify the factors associated with an elevated level of CVD risk separately for both sexes. We imputed missing values for the variables sedentary time (6.9%), body mass index (1.2%), systolic blood pressure (0.2%), diastolic blood pressure (0.2%) using the Hot-Deck method to include the cases as much as possible [20]. We calculated the CVD risk score according to the non-laboratory-based WHO CVD risk chart 2019 [10]. We performed descriptive analyses to report the background characteristics of the study population and the distribution of the CVD risk levels across the strata of the background characteristics. We then carried out univariable logistic regression to see the association of the elevated CVD risk ($\geq$ 10%) with the explanatory variables. However, we did not include age, sex, smoking status, body mass index, and hypertension in the univariable or multivariable analyses as those were used as the variables in the WHO CVD risk chart to calculate CVD risk. The multivariable logistic regression analysis was conducted with the variables with a p-value of $\leq$0.2 in the univariable logistics regression analysis [21]. Variance inflation factors (VIFs) were checked to assess multi-collinearity among variables. We reported crude and adjusted odds ratios with 95% confidence intervals and considered the factors statistically significant if the p-value was <0.05. Besides analyses with imputed missing values, we also performed complete case analysis and provided the results as S1 and S2 Tables.

## Ethical considerations

The Institutional Review Board (IRB) of the BRAC James P Grant School of Public Health, BRAC University, Dhaka, Bangladesh, provided the ethical approval of the FSNSP 2018–19 (IRB Reference number: 2018-020-IR). The research assistants obtained written informed consent from the respondents before data collection and measurements. The field coordinators met the community leaders and discussed the study purpose and procedure to ensure community consent.

## Results

We enrolled 30,003 participants of 6 different age groups from 17,323 households. For this analysis, we extracted data of 7,757 males and females aged from 40 years to 74 years, as the

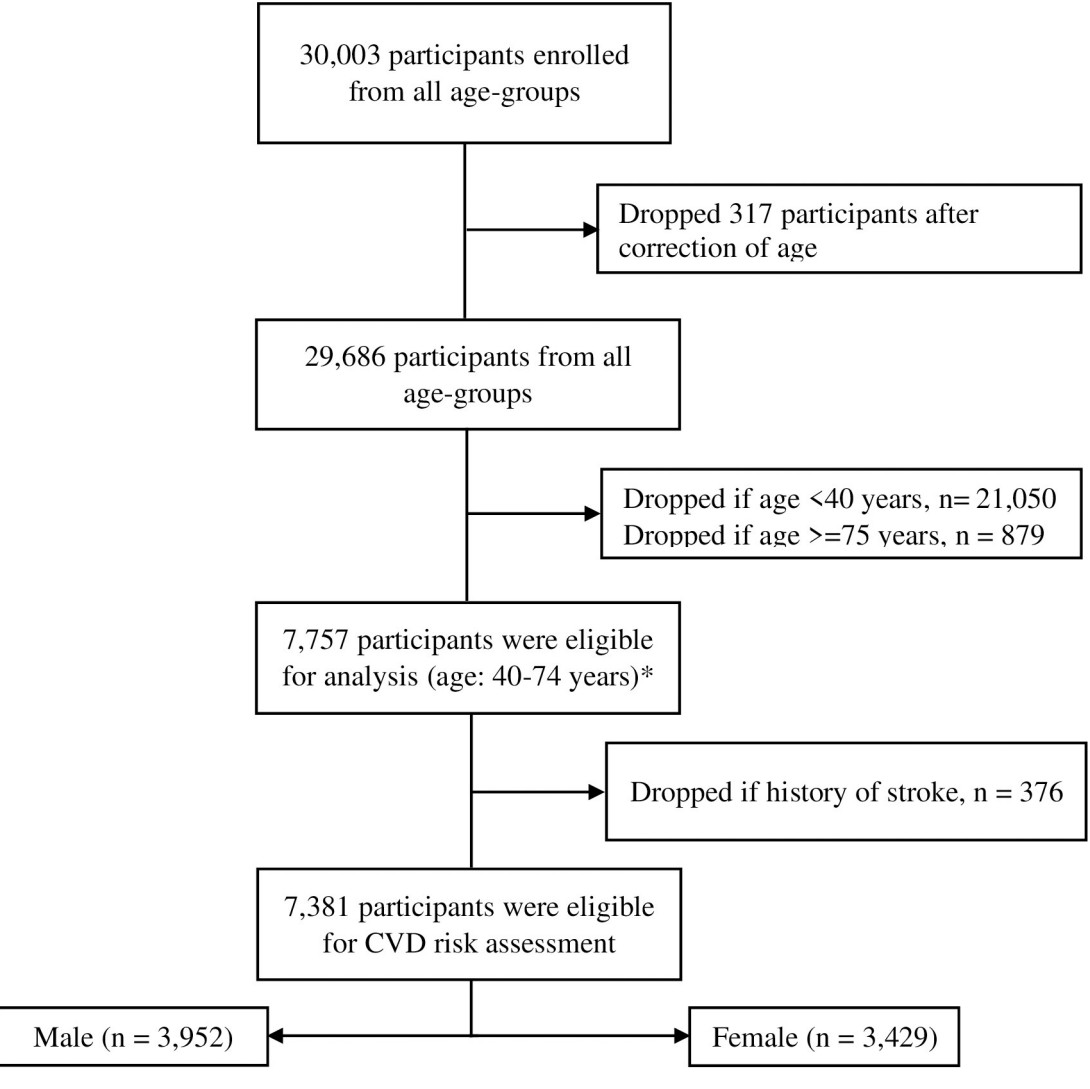

* WHO CVD Risk Assessment Chart applies only for persons aged 40-74 years

**Fig 1. Diagram showing the process of extracting eligible participants.**

WHO CVD risk chart is only applicable for this age group. We then dropped 376 participants who had a known history of CVD events (e.g., stroke), which provided us a sample size of 7,381 participants for CVD risk assessment using the WHO non-laboratory-based chart. Of the respondents, 3,429 (46%) were female. Fig 1 demonstrates how we extracted 7,381 participants eligible for the CVD risk assessment algorithm from the entire study population.

Table 1 describes some key sociodemographic, behavioral, and biological characteristics of the study participants. The median age with interquartile range was 58.6 (48.1–64.6) years and 60.2 (48.0–64.8) years for males and females, respectively. About three-fourths of the participants were from rural areas. About half of the males (45%) and about two-thirds of the females (66%) had no formal education. While 97% of the males were currently married, about half of the women (43%) were either divorced, separated, widowed, or never married at the time of the interview. Islam was the religion of 85% of the respondents. Insufficient physical activity was reported by 28% of the participants, whereas 88% reported inadequate consumption of

**Table 1. Sociodemographic, behavioral and biological characteristics of the study participants by sex.**

| | Overall (N = 7,381) | Male (N = 3,952) | Female (N = 3,429) | P-value (Chi^2)** |
|---|---|---|---|---|
| | n (%)* | n (%)* | n (%)* | |
| Age groups (years) | | | | 0.011 |
| 40–49 | 2,278 (30.9) | 1,219 (30.8) | 1,059 (30.9) | |
| 50–59 | 1,436 (19.5) | 818 (20.7) | 618 (18.0) | |
| 60–69 | 3,009 (40.8) | 1,587 (40.2) | 1,422 (41.5) | |
| 70–74 | 658 (8.9) | 328 (8.3) | 330 (9.6) | |
| Place of residence | | | | 0.061 |
| Rural | 5,355 (72.6) | 2,908 (73.6) | 2,447 (71.4) | |
| Non-slum urban | 1,226 (16.6) | 643 (16.3) | 583 (17.0) | |
| Slum | 800 (10.8) | 401 (10.1) | 399 (11.6) | |
| Educational status | | | | <0.001 |
| No formal education | 4,024 (54.5) | 1,771 (44.8) | 2,253 (65.7) | |
| 1–5 years | 935 (12.7) | 541 (13.7) | 394 (11.5) | |
| 6–10 years | 1,601 (21.7) | 1,006 (25.5) | 595 (17.4) | |
| >10 years | 821 (11.1) | 634 (16.0) | 187 (5.5) | |
| Household income | | | | <0.001 |
| Lowest (Q1) | 1,705 (23.1) | 811 (20.5) | 894 (26.1) | |
| Lower (Q2) | 1,453 (19.7) | 800 (20.2) | 653 (19.0) | |
| Middle (Q3) | 1,727 (23.4) | 964 (24.4) | 763 (22.3) | |
| Higher (Q4) | 1,049 (14.2) | 604 (15.3) | 445 (13.0) | |
| Highest(Q5) | 1,446 (19.6) | 772 (19.5) | 674 (19.7) | |
| Marital Status | | | | <0.001 |
| Currently married | 5,760 (78.0) | 3,818 (96.6) | 1,942 (56.6) | |
| Others£ | 1,621 (22.0) | 134 (3.4) | 1,487 (43.4) | |
| Religion | | | | 0.81 |
| Muslim | 6,276 (85.0) | 3,364 (85.1) | 2,912 (84.9) | |
| Others££ | 1,105 (15.0) | 588 (14.9) | 517 (15.1) | |
| Physical Activity | | | | 0.009 |
| > = 150 Minutes/week | 5,295 (71.7) | 2,785 (70.5) | 2,510 (73.2) | |
| <150 Minutes/week | 2,086 (28.3) | 1,167 (29.5) | 919 (26.8) | |
| Sedentary time per day | | | | <0.001 |
| < = 240 minutes | 2,721 (36.9) | 1,528 (38.7) | 1,193 (34.8) | |
| 241 to 360 minutes | 2,381 (32.3) | 1,175 (29.7) | 1,206 (35.2) | |
| >360 minutes | 2,279 (30.9) | 1,249 (31.6) | 1,030 (30.0) | |
| Fruits & Vegetables Consumption | | | | <0.001 |
| > = 5 servings/day | 913 (12.4) | 606 (15.3) | 307 (9.0) | |
| <5 servings/day | 6,468 (87.6) | 3,346 (84.7) | 3,122 (91.0) | |
| Current smoker | | | | <0.001 |
| No | 5,538 (75.0) | 2,179 (55.1) | 3,359 (98.0) | |
| Yes | 1,843 (25.0) | 1,773 (44.9) | 70 (2.0) | |
| Current smokeless tobacco user | | | | <0.001 |
| No | 4,138 (56.1) | 2,491 (63.0) | 1,647 (48.0) | |
| Yes | 3,243 (43.9) | 1,461 (37.0) | 1,782 (52.0) | |
| Self-reported diabetes | | | | <0.001 |
| No | 6,636 (89.9) | 3,632 (91.9) | 3,004 (87.6) | |
| Yes | 745 (10.1) | 320 (8.1) | 425 (12.4) | |
| Hypertension | | | | <0.001 |

(*Continued*)

**Table 1.** (Continued)

| | Overall (N = 7,381) | Male (N = 3,952) | Female (N = 3,429) | P-value (Chi^2)** |
|---|---|---|---|---|
| | n (%)* | n (%)* | n (%)* | |
| No | 4,456 (60.4) | 2,659 (67.3) | 1,797 (52.5) | |
| Yes | 2,917 (39.6) | 1,292 (32.7) | 1,625 (47.5) | |
| Body Mass Index (BMI) | | | | <0.001 |
| Normal | 3,203 (43.8) | 1,881 (47.9) | 1,322 (39.0) | |
| Underweight | 1,300 (17.8) | 739 (18.8) | 561 (16.6) | |
| Overweight | 2,053 (28.1) | 1,053 (26.8) | 1,000 (29.5) | |
| Obese | 761 (10.4) | 257 (6.5) | 504 (14.9) | |
| Waist Circumference | | | | <0.001 |
| Male: <90 cm/ Female: <80 cm | 4,744 (64.6) | 2,998 (76.1) | 1,746 (51.2) | |
| Male: > = 90 cm/ Female: > = 80 cm | 2,605 (35.4) | 941 (23.9) | 1,664 (48.8) | |

*Column percentage

**Chi^2 test between sex and the listed characteristics of the study participants.

£Never married, widows, divorced and separated.

££Hindu, Christian, Buddhist and others.

fruits and vegetables. About 31% of the participants spent at least six hours on a typical day, either sitting or reclining (other than sleep). Among the participants, 45% of the males and 2% of females reported as current smokers, whereas 37% of males and 52% of females reported as current users of smokeless tobacco. About one-third of the males (33%) and half of the females (48%) were hypertensive, while 8% of males and 12% of females had a history of diabetes. We did not find much difference between the males and females regarding the prevalence of underweight (19% vs. 17%) and overweight (27% vs. 30%). However, the prevalence of obesity among females was two-times higher than their male counterparts (15% vs. 7%).

Overall, the prevalence of very low, low, moderate, high, and very high CVD risk was 34.7%, 37.8%, 25.9%, 1.6%, and 0.1%, respectively. Moreover, the proportion of participants with moderate to very high CVD risk (henceforth 'elevated CVD risk' with a risk score ≥10%) was 33.7%, 20.4%, and 27.5% among the males, females, and both sexes, respectively. Table 2 provides the distribution of 10-years CVD risk according to the sociodemographic and other background characteristics. The proportion of participants with elevated CVD risk was higher among advanced age groups. For example, only 0.4% of the persons aged 40–49 years had elevated CVD risk, whereas the proportion of the same was 7.3% in 50–59 years, 41.9% in 60–69 years, and 100% among those aged 70+ years. Males, unmarried (never married, divorced, widowed, separated), non-slum urban residents, and the non-Muslims had a higher prevalence of elevated CVD risk compared to females, currently married, rural/slum residents, and Muslims, respectively. Fig 2 visualizes the pattern of the prevalence of elevated CVD risk, i.e., risk ≥10% by sex and age of the participants.

Tables 3 and 4 report univariable and multivariable logistic regression results for males and females, respectively. In univariable regression for males, higher household income, being unmarried (never married, divorced, widowed, separated), insufficient physical activity, more sedentary time, inadequate intake of fruits and vegetable, being a smokeless tobacco user, having self-reported diabetes, and higher waist circumference was positively associated with elevated CVD risk, whereas in females, all those factors except higher waist circumference were positively associated with an elevated level of CVD risk. However, in the multivariable logistic regression analysis, being unmarried (never married, divorced, widowed, separated),

**Table 2. Distribution of 10-years risk of fatal or non-fatal cardiovascular risk, according to background characteristics.**

| Variables | Very low to low (<10%) | Moderate (≥10% to <20%) | High to very high (≥20%) | P-value (Chi^2)** |
|---|---|---|---|---|
| | n (%)* | n (%)* | n (%)* | |
| Overall | 5,351 (72.5) | 1,908 (25.9) | 122 (1.7) | |
| Sex | | | | <0.001 |
| Male | 2,621 (66.3) | 1,236 (31.3) | 95 (2.4) | |
| Female | 2,730 (79.6) | 672 (19.6) | 27 (0.8) | |
| Age groups (years) | | | | <0.001 |
| 40–49 | 2,270 (99.7) | 8 (0.4) | 0 (0.0) | |
| 50–59 | 1,331 (92.7) | 103 (7.2) | 2 (0.1) | |
| 60–69 | 1,750 (58.2) | 1,220 (40.6) | 39 (1.3) | |
| 70–74 | 0 (0.0) | 577 (87.7) | 81 (12.3) | |
| Place of residence | | | | 0.324 |
| Rural | 3,904 (72.9) | 1,359 (25.4) | 92 (1.7) | |
| Non-slum urban | 863 (70.4) | 346 (28.2) | 17 (1.4) | |
| Slum | 584 (73.0) | 203 (25.4) | 13 (1.6) | |
| Educational status | | | | 0.321 |
| No formal education | 2,874 (71.4) | 1,083 (26.9) | 67 (1.7) | |
| 1–5 years | 689 (73.7) | 229 (24.5) | 17 (1.8) | |
| 6–10 years | 1,187 (74.1) | 392 (24.5) | 22 (1.4) | |
| >10 years | 601 (73.2) | 204 (24.9) | 16 (2.0) | |
| Household income | | | | 0.032 |
| Lowest (Q1) | 1,296 (76.0) | 385 (22.6) | 24 (1.4) | |
| Lower (Q2) | 1,040 (71.6) | 384 (26.4) | 29 (2.0) | |
| Middle (Q3) | 1,254 (72.6) | 444 (25.7) | 29 (1.7) | |
| Higher (Q4) | 739 (70.5) | 294 (28.0) | 16 (1.5) | |
| Highest(Q5) | 1,022 (70.7) | 400 (27.7) | 24 (1.7) | |
| Marital Status | | | | <0.001 |
| Currently married | 4,327 (75.1) | 1,346 (23.4) | 87 (1.5) | |
| Others£ | 1,024 (63.2) | 562 (34.7) | 35 (2.2) | |
| Religion | | | | 0.097 |
| Muslim | 4,577 (73.0) | 1,600 (25.5) | 99 (1.6) | |
| Others££ | 774 (70.1) | 308 (27.9) | 23 (2.1) | |
| Physical Activity | | | | <0.001 |
| > = 150 Minutes/week | 4,107 (77.6) | 1128 (21.3) | 60 (1.1) | |
| <150 Minutes/week | 1,244 (59.6) | 780 (37.4) | 62 (3.0) | |
| Sedentary time per day | | | | <0.001 |
| < = 240 minutes | 2,097 (77.1) | 594 (21.8) | 30 (1.1) | |
| 241 to 360 minutes | 1,705 (71.6) | 633 (26.6) | 43 (1.8) | |
| >360 minutes | 1,549 (68.0) | 681 (29.9) | 49 (2.2) | |
| Fruits & Vegetables Consumption | | | | 0.148 |
| > = 5 servings/day | 685 (75.0) | 212 (23.2) | 16 (1.8) | |
| <5 servings/day | 4,666 (72.1) | 1,696 (26.2) | 106 (1.6) | |
| Current smoker | | | | <0.001 |
| No | 4,262 (77.0) | 1,222 (22.1) | 54 (1.0) | |
| Yes | 1,089 (59.1) | 686 (37.2) | 68 (3.7) | |
| Current smokeless tobacco user | | | | 0.178 |
| No | 3,035 (73.3) | 1,036 (25.0) | 67 (1.6) | |
| Yes | 2,316 (71.4) | 872 (26.9) | 55 (1.7) | |

*(Continued)*

**Table 2.** (Continued)

| Variables | Very low to low (<10%) | Moderate (≥10% to <20%) | High to very high (≥20%) | P-value (Chi^2)** |
|---|---|---|---|---|
| | n (%)* | n (%)* | n (%)* | |
| Self-reported diabetes | | | | <0.001 |
| No | 4,862 (73.3) | 1,663 (25.1) | 111 (1.7) | |
| Yes | 489 (65.6) | 245 (32.9) | 11 (1.5) | |
| Hypertension | | | | <0.001 |
| No | 3,650 (81.9) | 804 (18.0) | 2 (0.0) | |
| Yes | 1694 (58.1) | 1,103 (37.8) | 120 (4.1) | |
| Body Mass Index (BMI) | | | | 0.042 |
| Normal | 2,299 (71.8) | 856 (26.7) | 48 (1.5) | |
| Underweight | 930 (71.5) | 347 (26.7) | 23 (1.8) | |
| Overweight | 1,515 (73.8) | 500 (24.4) | 38 (1.9) | |
| Obese | 587 (77.1) | 165 (21.7) | 9 (1.2) | |
| Waist Circumference | | | | 0.129 |
| Male: <90 cm/ Female: <80 cm | 3,430 (72.3) | 1,243 (26.2) | 71 (1.5) | |
| Male: > = 90 cm/ Female: > = 80 cm | 1,910 (73.3) | 644 (24.7) | 51 (2.0) | |

*Row percentage

**Chi^2 test between risk categories and the listed characteristics of the study participants.

£Never married, widows, divorced and separated.

££Hindu, Christian, Buddhist and others except Muslims.

insufficient physical activity, smokeless tobacco use and having self-reported diabetes in both sexes; higher household income and higher sedentary time among only males; and being slum dwellers and non-Muslim among only females were positively associated with elevated CVD

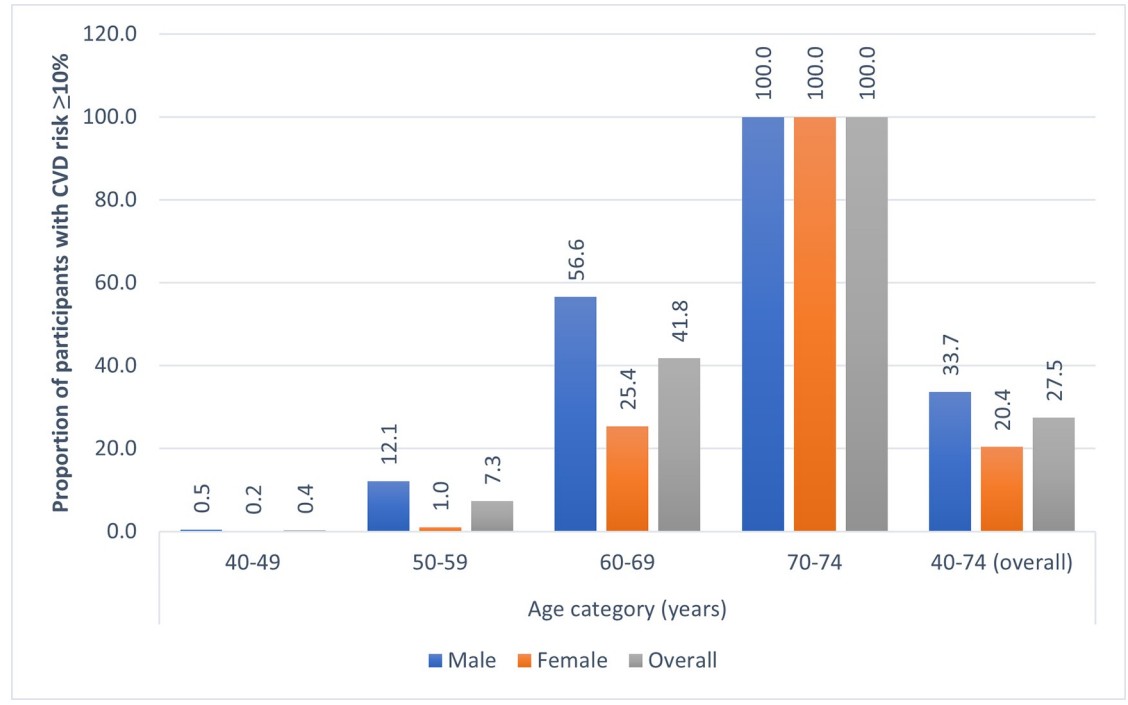

**Fig 2. The prevalence of elevated CVD risk (CVD risk ≥10%) by sex and age of the participants.**

**Table 3. Univariable and multivariable logistic regression results with potential determinants of elevated CVD risk ≥10% (for males)\*.**

| Variables | Univariable Logistic Regression | | | Multivariable Logistic Regression\*\* | | |
|---|---|---|---|---|---|---|
| | COR | 95% CI | P value | AOR | 95% CI | P-value |
| Place of residence | | | | | | |
| Rural | Ref | | | Ref | | |
| Non-slum urban | 1.03 | 0.86, 1.23 | 0.771 | NA | NA | NA |
| Slum | 1.06 | 0.85, 1.31 | 0.632 | NA | NA | NA |
| Educational status | | | | | | |
| No formal education | Ref | | | Ref | | |
| 1–5 years | 0.96 | 0.78, 1.18 | 0.705 | 0.89 | 0.72, 1.09 | 0.258 |
| 6–10 years | 0.89 | 0.75, 1.04 | 0.150 | 0.79 | 0.66, 0.94 | 0.007 |
| >10 years | 0.89 | 0.74, 1.08 | 0.244 | 0.62 | 0.50, 0.78 | <0.001 |
| Household income | | | | | | |
| Lowest (Q1) | Ref | | | Ref | | |
| Lower (Q2) | 1.18 | 0.95, 1.45 | 0.128 | 1.17 | 0.95, 1.45 | 0.144 |
| Middle (Q3) | 1.04 | 0.85, 1.28 | 0.692 | 1.09 | 0.88, 1.34 | 0.435 |
| Higher (Q4) | 1.35 | 1.08, 1.68 | 0.009 | 1.34 | 1.06, 1.69 | 0.013 |
| Highest(Q5) | 1.34 | 1.08, 1.65 | 0.007 | 1.33 | 1.06, 1.67 | 0.013 |
| Marital Status | | | | | | |
| Currently married | Ref | | | Ref | | |
| Others\*\*\* | 3.14 | 2.21, 4.47 | <0.001 | 2.88 | 2.00, 4.14 | <0.001 |
| Religion | | | | | | |
| Muslim | Ref | | | Ref | | |
| Others\*\*\*\* | 1.09 | 0.91, 1.31 | 0.346 | NA | NA | NA |
| Physical Activity | | | | | | |
| > = 150 Minutes/week | Ref | | | Ref | | |
| <150 Minutes/week | 1.92 | 1.66, 2.21 | <0.001 | 1.77 | 1.52, 2.07 | <0.001 |
| Sedentary time per day | | | | | | |
| < = 240 minutes | Ref | | | Ref | | |
| 241 to 360 minutes | 1.47 | 1.25, 1.72 | <0.001 | 1.34 | 1.13, 1.58 | 0.001 |
| >360 minutes | 1.50 | 1.27, 1.75 | <0.001 | 1.20 | 1.01, 1.43 | 0.035 |
| Fruits & Vegetables Consumption | | | | | | |
| > = 5 servings/day | Ref | | | Ref | | |
| <5 servings/day | 1.21 | 1.00, 1.46 | 0.049 | 1.06 | 0.87, 1.28 | 0.570 |
| Current smokeless tobacco user | | | | | | |
| No | Ref | | | Ref | | |
| Yes | 1.18 | 1.03, 1.36 | 0.015 | 1.21 | 1.06, 1.40 | 0.007 |
| Self-reported diabetes | | | | | | |
| No | Ref | | | Ref | | |
| Yes | 1.66 | 1.32, 2.09 | <0.001 | 1.55 | 1.21, 1.98 | 0.001 |
| Waist Circumference | | | | | | |
| Male: <90 cm/ Female: <80 cm | Ref | | | Ref | | |
| Male: > = 90 cm/ Female: > = 80 cm | 1.26 | 1.09, 1.47 | 0.003 | 1.15 | 0.97, 1.36 | 0.110 |

\*The regression analysis outcome was elevated CVD risk (risk of CVD events in 10 years is > = 10%: No = 0/Yes = 1).

\*\*Adjusted for educational status, household income, marital status, physical activity, sedentary time per day, fruits and vegetable consumption, current user of smokeless tobacco, self-reported diabetes, and waist circumference (age, sex, smoking status, BMI, and hypertension were used to calculated CVD risk, i.e., the outcome).

\*\*\*Never married, widows, divorced and separated.

\*\*\*\*Hindu, Christian, Buddhist, and others.

CI: Confidence Interval; COR: Crude Odds Ratio; AOR: Adjusted Odds Ratio; Ref: Reference category.

NA: Not applicable; these variables were not included in the adjusted analysis as these were dropped due to significance level was >0.2 in the crude analysis.

**Table 4. Univariable and multivariable logistic regression results with potential determinants of elevated CVD risk ≥10% (for females)\*.**

| Variables | Univariable Logistic Regression | | | Multivariable Logistic Regression\*\* | | |
|---|---|---|---|---|---|---|
| | COR | 95% CI | P value | AOR | 95% CI | P-value |
| Place of residence | | | | | | |
| Rural | Ref | | | Ref | | |
| Non-slum urban | 1.35 | 1.09, 1.67 | 0.006 | 1.08 | 0.84, 1.39 | 0.554 |
| Slum | 0.99 | 0.75, 1.29 | 0.912 | 0.66 | 0.49, 0.88 | 0.005 |
| Educational status | | | | | | |
| No formal education | Ref | | | Ref | | |
| 1–5 years | 0.60 | 0.45, 0.81 | 0.001 | 0.68 | 0.50, 0.93 | 0.015 |
| 6–10 years | 0.58 | 0.45, 0.74 | <0.001 | 0.60 | 0.46, 0.79 | <0.001 |
| >10 years | 0.28 | 0.16, 0.48 | <0.001 | 0.33 | 0.18, 0.59 | <0.001 |
| Household income | | | | | | |
| Lowest (Q1) | Ref | | | Ref | | |
| Lower (Q2) | 1.24 | 0.97, 1.60 | 0.090 | 1.15 | 0.88, 1.51 | 0.315 |
| Middle (Q3) | 1.31 | 1.03, 1.66 | 0.030 | 1.20 | 0.92, 1.56 | 0.170 |
| Higher (Q4) | 1.08 | 0.81, 1.45 | 0.593 | 1.05 | 0.77, 1.44 | 0.753 |
| Highest(Q5) | 1.17 | 0.91, 1.51 | 0.213 | 1.27 | 0.95, 1.69 | 0.108 |
| Marital Status | | | | | | |
| Currently married | Ref | | | Ref | | |
| Others\*\*\* | 5.11 | 4.24, 6.15 | <0.001 | 3.96 | 3.26, 4.82 | <0.001 |
| Religion | | | | | | |
| Muslim | Ref | | | Ref | | |
| Others\*\*\*\* | 1.27 | 1.01, 1.58 | 0.037 | 1.50 | 1.17, 1.92 | 0.001 |
| Physical Activity | | | | | | |
| > = 150 Minutes/week | Ref | | | Ref | | |
| <150 Minutes/week | 3.15 | 2.65, 3.75 | <0.001 | 2.29 | 1.88, 2.78 | <0.001 |
| Sedentary time per day | | | | | | |
| < = 240 minutes | Ref | | | Ref | | |
| 241 to 360 minutes | 1.34 | 1.09, 1.65 | 0.006 | 1.12 | 0.89, 1.40 | 0.340 |
| >360 minutes | 1.83 | 1.48, 2.25 | <0.001 | 1.22 | 0.97, 1.54 | 0.087 |
| Fruits & Vegetables Consumption | | | | | | |
| > = 5 servings/day | Ref | | | Ref | | |
| <5 servings/day | 1.54 | 1.11, 2.14 | 0.010 | 1.12 | 0.79, 1.60 | 0.517 |
| Current smokeless tobacco user | | | | | | |
| No | Ref | | | Ref | | |
| Yes | 1.30 | 1.10, 1.54 | 0.002 | 1.23 | 1.02, 1.48 | 0.032 |
| Self-reported diabetes | | | | | | |
| No | Ref | | | Ref | | |
| Yes | 1.49 | 1.18, 1.89 | 0.001 | 1.38 | 1.06, 1.80 | 0.016 |
| Waist Circumference | | | | | | |
| Male: <90 cm/ Female: <80 cm | Ref | | | Ref | | |
| Male: > = 90 cm/ Female: > = 80 cm | 1.04 | 0.88, 1.23 | 0.623 | NA | NA | NA |

\*The regression analysis outcome was elevated CVD risk (risk of CVD events in 10 years is > = 10%: No = 0/Yes = 1).

\*\*Adjusted for educational status, household income, marital status, physical activity, sedentary time per day, fruits and vegetable consumption, current user of smokeless tobacco, self-reported diabetes (age, sex, smoking status, BMI, and hypertension were used to calculate CVD risk i.e., the outcome).

\*\*\*Never married, widows, divorced and separated.

\*\*\*\*Hindu, Christian, Buddhist, and others.

CI: Confidence Interval; COR: Crude Odds Ratio; AOR: Adjusted Odds Ratio; Ref: Reference category.

NA: Not applicable; these variables were not included in the adjusted analysis as these were dropped due to significance level was >0.2 in the crude analysis.

risk. Higher educational status was negatively associated with elevated CVD risk in both sexes in multivariable logistic regression analysis.

Moreover, in the adjusted analysis, gradual decline of elevated CVD risk with the level of education suggesting a dose-response relationship among males with aORs (95% CI): 0.89 (0.72–1.09), 0.79 (0.66–0.94), and 0.62 (0.50–0.78) and among females with aORs (95% CI): 0.68 (0.50–0.93), 0.60 (0.46–0.79) and 0.33 (0.18–0.59) for education levels 'up to five years', 'ten years,' and 'more than ten years', respectively, compared to the participants reporting no formal education. Both males and females with insufficient physical activity had about two times higher odds of having elevated CVD risk compared to those who were reportedly physically active. The odds of elevated CVD risk were threefold higher among males (aOR: 2.88, 95% CI: 2.00 to 4.14) and fourfold higher among females (aOR: 3.96, 95% CI: 3.26 to 4.82) who were divorced, widowed, separated, or never married during the survey. Self-reported diabetes also increased odds of elevated CVD risk by 1.5 times in males (95% CI: 1.21 to 1.98) and 1.38 times in females (95% CI: 1.06 to 1.80). Moreover, females living in slums had lower odds (aOR: 0.66, 95% CI: 0.49 to 0.88) of elevated CVD risk.

## Discussion

There is evidence that in most cases, CVD events such as myocardial infarction, stroke do not occur due to a single risk factor. Instead, it precipitates from the holistic effect of multiple risk factors [22]. In general, the evaluation of cardiovascular disease risk by general physicians is limited [23]. In this study, we aimed to estimate the 10-years risk of fatal or non-fatal CVD events among the males and females of Bangladesh by using the WHO non-laboratory-based CVD risk chart. To the best of our knowledge, this is the first study in Bangladesh, in which we calculated CVD risk with nationally representative data. We found that 27.5% of the study population (aged 40 to 74 years), i.e., one in every four participants, had elevated risk (CVD risk ≥10%) of a fatal or non-fatal CVD events in the next 10 years of the survey [Fig 2]. This prevalence of CVD risk is higher than a previous study in Bangladesh carried out by Fatema. et al., in which the rate was 20.2% [9]. Another South Indian study reported the rate as 14% [22]. The possible reason might be that both of these studies were conducted in rural areas in 2011–2012 and the difference in age ranges (31–74 years in Fatema. Et al.). Another cross-sectional study in Nepal also reported a lower proportion (13.6%) of elevated CVD risk though the study was not nationally representative [24]. In our study, the proportion of the population with a high to very high risk of CVD (CVD risk ≥20%) was 1.7%. This proportion was much lower than the findings from other studies in Bangladesh (11%) [9], Nepal (4.3%) [24], rural south India (male 4.5% and female 5.3%) [22]. The lower proportion of high to very high CVD risk in our study might be due to the difference between the charts we used and is in alignment with the recommendations of the WHO CVD Risk Chart Working Group who reported moderate agreement between risk predictions using laboratory and non-laboratory based charts [10]. Despite this limitation of the non-laboratory-based chart, we can say that in the areas where laboratory testing is not convenient or unavailable, this low-cost, low-skill approach can simplify the CVD risk assessment of the people aged 40–74 years.

In this population, elevated CVD risk was higher among males than females (p <0.001). While about 31% of the males were in the moderate risk category, this proportion was about 20% in the case of females. The proportion of males in the high to the very high-risk category (CVD risk ≥20%) was three times higher compared to females. This sex difference in the prevalence of CVD risk is also evident in other studies [22,24]. However, in contrast to our findings, at least one study reported mixed findings, i.e., males had a higher proportion in the moderate-risk group whereas, in the high-risk group, females were in a higher proportion [9].

This difference may be due to the cause that men are at risk of developing cardiovascular diseases 7 to 10 years earlier compared to women [25]. Moreover, the effect of endogenous estrogen during the reproductive period delays the onset of CVD in women [25]. Evidence from the Women's Ischemia Syndrome Evaluation (WISE) study showed that the risk of coronary arterial diseases is sevenfold higher in women with endogenous estrogen deficiency [26].

CVD risk was negatively associated with the education level among males and females. For those with more than 10 years of education, the odds of elevated CVD risk (CVD risk ≥10%) were lower by one-third in males and two-thirds in females than those with no formal education. Studies in Nepal [24] and Austria [27] found similar associations between education level and elevated CVD risk. Education might be a determining factor in lifestyle modification [28]. A study from Chicago, IL, USA, reported that knowledge of risk factors could motivate people to change their risky behavior [29]. In our study, marital status was also found associated with elevated CVD risk. Those who were divorced, separated, widowed, or never married during the interview had higher odds of elevated CVD risk in 10 years. Findings from a prospective cohort study also suggest the same [30]. Divorced, separated, widowed, or never-married persons may have reduced social support and reduced motivation to lead an active and healthy lifestyle [30].

Physical inactivity, smokeless tobacco use were also found associated with higher odds of elevated CVD risk among both sexes in this population. However, high household income and high sedentary time were significantly associated with CVD risk only among males. These are established risk factors of cardiovascular diseases, and these findings are consistent with several studies in Bangladesh and elsewhere [1,31]. Insufficient physical activity and other behavioral and clinical risk factors of NCDs, including overweight, high blood pressure, tobacco, and high blood sugar levels, are interrelated and can influence each other [7].

The use of the WHO non-laboratory-based CVD risk prediction chart can help in the early detection of individuals with an elevated or high to very high risk of having fatal or non-fatal CVD events and can supplement the prevention and management programs of CVD in resource-poor settings. It can be useful, especially in resource-poor settings, where the health facilities are not equipped to detect blood sugar, and serum cholesterol levels as the WHO laboratory-based CVD risk chart needs these data. Prediction of CVD risk using the non-laboratory-based chart can also reduce the proportion of population needing pharmacological intervention as the people with low to moderate risk of CVD can be managed by an integrated approach to lifestyle changes regular follow-up. Health care professionals, particularly those working in the primary health care system in resource-poor settings, should be trained on this chart so that they can contribute to the prevention and control of CVD and related diseases by implementing screening, counseling, management, and referral.

## Strengths and limitations

In Bangladesh, few studies have reported the CVD risk based on the WHO CVD risk chart. However, to the best of our knowledge, this is the first-ever study in Bangladesh reporting CVD risk using the non-laboratory-based WHO CVD chart as well as using nationally representative data. Our study also had some limitations, and one of them is sampling challenges during the implementation of the study. We had to drop seven pre-selected rural clusters due to administrative and financial constraints, which might affect the overall representativeness of the study. To calculate the CVD risk using the aforementioned chart, one needs to drop the participants if s/he has a history of CVD events such as heart attack or stroke. In our study, although we have collected data on the history of heart diseases, data on the history of myocardial infarction (MI) or heart attack could not be separated from that. So, we only dropped the

participants who had a history of stroke but could not drop those who had a history of MI or heart attack, which might have influenced the prevalence of elevated CVD risk. Also, this is a cross-sectional study, and therefore, the temporality of the associations revealed in this study cannot be established. To include the highest number of cases in this analysis, we imputed missing values for specific variables. Besides, due to lack of data, some possible confounders such as salt consumption, genetic factors, family history could not be taken into account. We, therefore, emphasize the importance of further research on the determinants of elevated CVD risk in this population using both laboratory-based and non-laboratory-based CVD risk charts.

## Conclusion

One in every four Bangladeshi adults had elevated levels of CVD risk, and males are at higher risk of experiencing CVD events. Non-laboratory-based risk prediction charts can be effectively used in low resource settings in Bangladesh and elsewhere. The government of Bangladesh should train the primary health care workers on the use of non-laboratory-based charts, relevant control, and management options in places where laboratory tests are not available.

## Supporting information

**S1 Table. Complete case analysis: Univariable and multivariable logistic regression results with potential determinants of elevated CVD risk ≥10% (for males).**
(DOCX)

**S2 Table. Complete case analysis: Univariable and multivariable logistic regression results with potential determinants of elevated CVD risk ≥10% (for females).**
(DOCX)

**S1 Data. Dataset in Stata format.** This is the final dataset we used to produce all the tables and figures used in the manuscript.
(DTA)

**S1 File. Questionnaire in English language.**
(PDF)

**S2 File. Questionnaire in Bengali language.**
(PDF)

## Acknowledgments

We acknowledge all the participants of the study and their family members, Research Assistants, Field Supervisors, community leaders, all the administrative and accounts staff of the BRAC James P Grant School of Public Health and Institute of Public Health and Nutrition, members of the Technical Advisory Committee, and the local administrators for their support during this work.

## Author Contributions

**Conceptualization:** Abu Abdullah Mohammad Hanif.

**Data curation:** Abu Abdullah Mohammad Hanif, Md. Mokbul Hossain.

**Formal analysis:** Abu Abdullah Mohammad Hanif.

**Funding acquisition:** Malay Kanti Mridha.

**Investigation:** Mehedi Hasan, Md Showkat Ali Khan, Abu Ahmed Shamim, Samir Kanti Sarker, Dipak Kumar Mitra, Malay Kanti Mridha.

**Methodology:** Mehedi Hasan, Md Showkat Ali Khan, Abu Ahmed Shamim, Samir Kanti Sarker, Dipak Kumar Mitra, Malay Kanti Mridha.

**Project administration:** Abu Abdullah Mohammad Hanif, Mehedi Hasan, Md Showkat Ali Khan, Md. Mokbul Hossain, Moyazzam Hossaine, Mohammad Aman Ullah, Samir Kanti Sarker, S. M Mustafizur Rahman, Md Mofijul Islam Bulbul, Malay Kanti Mridha.

**Resources:** S. M Mustafizur Rahman.

**Supervision:** Abu Abdullah Mohammad Hanif, Md Showkat Ali Khan, Md. Mokbul Hossain, Moyazzam Hossaine.

**Writing – original draft:** Abu Abdullah Mohammad Hanif.

**Writing – review & editing:** Abu Abdullah Mohammad Hanif, Mehedi Hasan, Malay Kanti Mridha.

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
