## [Decision Letter · Decision Letter 0]

1 Sep 2020

PONE-D-20-10140

Ten-years cardiovascular risk among the Bangladeshi population using non-laboratory-based risk chart of the World Health Organization: findings from a nationally representative survey

PLOS ONE

Dear Dr. Hanif,

Thank you for submitting your manuscript to PLOS ONE. After careful consideration, we feel that it has merit but does not fully meet PLOS ONE’s publication criteria as it currently stands. Therefore, we invite you to submit a revised version of the manuscript that addresses the points raised during the review process.

We look forward to receiving your revised manuscript.

Kind regards,

Giuseppe Vergaro, M.D., Ph.D.

Academic Editor

PLOS ONE

Journal Requirements:

"Some of the representatives of the Ministry of Health and Family Welfare, who

approved the funding of the study were involved with the Technical Advisory Group.

Though they had opinions about some maternal and child health and nutrition

indicators, they did not have any role in the design, conduct, data analysis, and

manuscript writing of the adolescent component of the study."

Reviewer's Responses to Questions

**Comments to the Author**

1. Is the manuscript technically sound, and do the data support the conclusions?

Reviewer #1: Yes

Reviewer #2: Yes

2. Has the statistical analysis been performed appropriately and rigorously? 

Reviewer #1: No

Reviewer #2: Yes

3. Have the authors made all data underlying the findings in their manuscript fully available?

Reviewer #1: Yes

Reviewer #2: Yes

4. Is the manuscript presented in an intelligible fashion and written in standard English?

Reviewer #1: Yes

Reviewer #2: Yes

5. Review Comments to the Author

Reviewer #1: In this paper, Abu Abdullah Mohammad Hanif and colleagues have addressed the estimation of CVD in Bangladeshi population based on non-laboratory-based charts.

This paper explores the topic of estimation and prevention of CVD in low income countries and it is therefore of interest for publication. It is a well displayed work with reasonable methodological section and interesting results.

I have nonetheless several issues to be addressed

Methods section

- In the “sample size and sampling techniques” section the Authors state that a sample size of 62 individuals from each cluster for each group was sampled. However, the age groups should be defined in the Methods section.

- In the Statistical analysis section the Authors state that they imputed missing values. However, imputation analysis might affect the results. A per protocol assessment of the results should also be performed.

- In the Statistical analysis section, the Authors have stated to have performed a bivariable logistic regression analysis. The variables enlisted should be declared. Why did the Authors choose a bivariable approach rather than a two step univariable-multivariable logistic regression analysis?

- In the variables considered to be relevant for CVD risk, family history is missing. The Authors further address the issue of genetics in the limitations section of the manuscript; however, family history is a relatively easy information to acquire. Can the Authors comment on this issue and add the family history data, if available?

Results section

- A CONSORT diagram would help the reader understanding the estimation of the population addressed, going from 30005 participant enrolled to 7757 actual data.

- The percentage 189% in line 294 is wrong and should be revised.

- The sentence in lines 294-296 is hard to understand and should be rephrased.

- In Table 1, the Authors enlist the general sociodemographic, behavioral and biological characteristics of the population. However, an analysis based on sex has also being carried on. Why did the Authors choose to stratify the population according to sex rather than other variables included in the WHO chart (i.e. Blood pressure)? The p-values addressed in the table are for multivariable comparisons? If so, sub-analyses for significantly different variables should also be performed.

- A figure on the prevalence of CVD and the prevalence of a risk>10% can help the reader focus on the main result of the paper.

- A Table assessing each variable of the WHO chart per risk category should also be performed to understand which risk factors account for the general CVD risk and on which it is possible to intervene (for example, diabetes or blood pressure rather than sex).

- What are the multivariable analysis adjusted for? Which is the covariate? Why did the Authors decide to stratify the population according to gender?

Discussion

- The sentence in lines 352-353 in difficult to understand and should be rephrased.

- Did the Authors consider to perform a lab-based analysis in a subgroup of their population? This could have validated their non-lab-based approach. Otherwise, the conclusions contained in lines 363-365 are too strong and should be rephrased.

- The paragraph between lines 389-396 is too general. Consider rephrasing.

- The intervention on CVD by primary physicians based on non-lab-based charts can only be hypothesized with this study, as follow up data were not collected. Therefore, the conclusions driven in lines 401-404 should be referred to as hypothetical.

References

- Ref 6,7, 15 and 19 are incomplete.

Reviewer #2: In the present paper, Abu Abdullah Mohammad Hanif and Colleagues aimed to investigate the role of a non-laboratory based cardiovascular disease (CVD) risk chart, recently developed by the World Health Organization (WHO)1, in assessing the prevalence of ten years CVD burdenamong Bangladeshi population aged 40-74 years.Parameters used in the risk algorithm were age, sex, smoking status, body mass index and systolic bloodpressure.They obtain six groups based on the calculated risk score:no risk (<5%), low risk (5% to <10%), moderate risk (10% to <20%), high risk (20% to <30%) and very high risk (≥30%).Then Authors listed potential factors associated with elevated CVD risk (defined as CVD risk ≥10%) among the subjects studied,based on literature review andconsidering data availability from this survey.Sociodemographic variables included place of residence (rural, non-slum urban, and slum), education, household income, marital status and religion; among the behavioralvariables were physical activity, sedentary time,fruits and vegetable consumption, and smokeless tobacco consumption status.So the study deals with an interestingissue concerning resource optimization topic; in low income countriesanon-laboratory based chart couldhelp the health care workers in screening, primary management counseling and referral of patients at higher risk of cardiovascular eventsto further evaluation. On the other hand, there aresome methodological limitations that could weak the strength of these findings,in particular data accuracy and poor method description.

Major points

-In the method session about sample size and sampling techniques,Authors have selectedpatients according to origin area: rural, urban and slum zone. For each cluster research assistants listed households based onage groups; could author clarify age range and better defined it (year of age or age range as listed in table 1?). Moreover,it could be interesting screening patients also by genderin order to have balanced groups of the same age

-In the method session authorsrecorded “self-reported diabetes”if subjects had declared this comorbidity; it could be useful regarding the elevated risk of CVD in this cluster of patients also checking old blood chemistry exams, when available.

-Method session need furtherexplanation; authors refer toa non-laboratorybasedCVD risk chart, recently developed by the World Health Organization1; but in the aforementioned model were included participants without a known baseline history of cardiovascular disease; this aspect isnot clarified in the study. Moreover,there are a lot of relevant parameters that are associated to an elevated risk of CVD, such as family history, chronic renal disease, carotid atherosclerosis,that have not been reported. All these variables are easily obtainable through an interview and have a significantly weight in the selection of clusters of subjects at higher risk of adverse events.

-In the “explanatory variables” session Authors defined outcome variable as elevated CVD risk if the total CVD risk score was ≥10% and CVD risk if the risk score was <10%; how they establish this cut-off?This choice needs further explanation.

-In the Statistical analysis section Authors carry out bivariate logistic regression analysis, but they don’t itemize thevariables sorted. Why did Authors employ a bivariable approach rather than a two-step univariable-multivariable logistic regression analysis?

-In the result section Authors state that they extracted data of 7,757 males and females aged from 40 to 74 years old as the WHO CVD risk chart is only applicable for this age groups, please clarify this content, since in the referral model1 were included participants aged 40-80 years old.

-An explicative graphic of the study design could be helpful in evaluating patient screening from the initial 17,323 households to the effective 7,757 subjects studied. 

-In Table 1, are p-values referred tomultivariable comparisons? Probably in this situation asub-analysisfor significantly different variables should be performed.

-Could authors explain why in table 2 they considered only sociodemographic characteristics? -All results are itemizedin fourtables;this presentation method could make occult some relevant data; afigure about the prevalence of CVD in the different risk categories could focus on thetopic of the paper.

-Could authors explain whatare the multivariable analysis adjusted for and which is the covariate?

-In the discussion session Authors state “this analysis revealed that without even any laboratory procedures, the primary health care worker would be able to assess the CVD risk of the population and initiate management/referral procedure following national protocols for the CVD risk”(lines 363-365); without follow-up data and in the absence of a comparison witha lab-based analysis (also in a subgroup of their sample size) they could not sentencing this content.

-In the discussion session the assertion: “insufficient physical activity may cause overweight, high blood pressure, and increase the level of bad cholesterol; tobacco can damage and narrow the blood vessels with harmful substances and, high blood sugar levels can damage the blood vessels, and all of these eventually cause CVDs”(lines 393-396) is too generic; please reward.

-Finally,considerationsabout the usefulness of a nonlab-based algorithm adopted byphysicians for implementing subsequent prevention and management options (line 397-399) can only be hypothesized, since follow-up data are missing. 

Minor points

-Please correct line 294 where we read 189% vs. 16%

-References are to be corrected 

-Please grammatically correct line 295 where probably verb is missing

-Please correct line 317: “,)”

-Please correct line 339: a bracket is missing

References

1) Kaptoge S, Pennells L, BacquerDD, Cooney MT, Kavousi M, Stevens G, et al. World Health Organization cardiovascular disease risk charts: revised models to estimate risk in 21 global regions. The Lancet Global Health. 2019;7(10):e1332-e45

6. PLOS authors have the option to publish the peer review history of their article (what does this mean?). If published, this will include your full peer review and any attached files.

Reviewer #1: No

Reviewer #2: No

---

## [Author Response · Author response to Decision Letter 0]

24 Feb 2021

PONE-D-20-10140

Ten-years cardiovascular risk among the Bangladeshi population using non-laboratory-based risk chart of the World Health Organization: findings from a nationally representative survey

Many thanks to the respected reviewers for reviewing the manuscript thoroughly and providing their valuable insights. We apologized that due to some unavoidable circumstances we could not submit the revised manuscript in timely fashion. We hope that the respected reviewers will accept our apology. 

Below the responses from the authors of this manuscript for the comments provided by the reviewer: 

5. Review Comments to the Author

Reviewer #1: In this paper, Abu Abdullah Mohammad Hanif and colleagues have addressed the estimation of CVD in Bangladeshi population based on non-laboratory-based charts.

This paper explores the topic of estimation and prevention of CVD in low income countries and it is therefore of interest for publication. It is a well displayed work with reasonable methodological section and interesting results.

I have nonetheless several issues to be addressed

Methods section

- In the “sample size and sampling techniques” section the Authors state that a sample size of 62 individuals from each cluster for each group was sampled. However, the age groups should be defined in the Methods section.

Response: We revised the paragraph and added detail description of sample size selection in each of the age groups. Please, see line numbers: 162-164

- In the Statistical analysis section the Authors state that they imputed missing values. However, imputation analysis might affect the results. A per protocol assessment of the results should also be performed.

Response: We performed a per-protocol analysis and listed the results of both univariable and multivariable logistic regression in supplementary tables. However, it is pertinent to mention that we did not find any important difference in the results of two analyses. 

- In the Statistical analysis section, the Authors have stated to have performed a bivariable logistic regression analysis. The variables enlisted should be declared. Why did the Authors choose a bivariable approach rather than a two-step univariable-multivariable logistic regression analysis?

Response: We added a footnote mentioning the variables we adjusted for in the multivariable logistic regression analysis. 

Regarding variables selection process, we indeed used univariable logistic regression to select variable for the multivariable logistic regression process. We corrected the texts accordingly. 

- In the variables considered to be relevant for CVD risk, family history is missing. The Authors further address the issue of genetics in the limitations section of the manuscript; however, family history is a relatively easy information to acquire. Can the Authors comment on this issue and add the family history data, if available?

Response: As mentioned in the method section of the paper, we collected data used in this analysis as part of a nationwide nutrition surveillance and due to limited resources, we could not collect family history of cardiovascular diseases along with some other important variables. We mentioned this as a weakness in the paper. Please, see the line numbers: 465 in the revised manuscript. However, we have plan to address this issue in future studies. 

Results section

- A CONSORT diagram would help the reader understanding the estimation of the population addressed, going from 30005 participant enrolled to 7757 actual data.

Response: We made a CONSORT flowchart demonstrating the process and added as Figure 1.

- The percentage 189% in line 294 is wrong and should be revised.

Response: Noted with thanks. We corrected the manuscript. Please, see line numbers: 314

- The sentence in lines 294-296 is hard to understand and should be rephrased.

Response: Noted with thanks. We revised the sentence. Please, see line numbers: 313-315.

- In Table 1, the Authors enlist the general sociodemographic, behavioral and biological characteristics of the population. However, an analysis based on sex has also being carried on. Why did the Authors choose to stratify the population according to sex rather than other variables included in the WHO chart (i.e. Blood pressure)? The p-values addressed in the table are for multivariable comparisons? If so, sub-analyses for significantly different variables should also be performed.

Response: We attempted to differentiate the entire analysis by sex considering the vulnerability, and poor health-care seeking practices of women in Bangladesh and similar settings to facilitate the policymakers focus on health-care of both women and men while designing and implementing the healthcare program to address NCDs, especially cardiovascular diseases. 

The p-values given on the table-1 were NOT from multivariable comparisons, rather just showing the difference between the male female proportion of the participants for each of the background characteristics. However, to eliminate confusion, we dropped the p-values from table and provided the revised table. 

- A figure on the prevalence of CVD and the prevalence of a risk>10% can help the reader focus on the main result of the paper.

Response: We provided a figure showing the prevalence of CVD risk >10% categorized by sex and ages and added as Figure 2. 

- A Table assessing each variable of the WHO chart per risk category should also be performed to understand which risk factors account for the general CVD risk and on which it is possible to intervene (for example, diabetes or blood pressure rather than sex).

Response: We added rows in Table 2 with all the variables used in the WHO CVD risk chart and showed the prevalence of risk category. 

- What are the multivariable analysis adjusted for? Which is the covariate? Why did the Authors decide to stratify the population according to gender?

Response: We used univariable-multivariable approach to select variables and conducted multivariable logistic regression. First, we carried out univariable logistic regression to see the association of the elevated CVD risk (CVD risk ≥10%) with the explanatory variables. The multivariable logistic regression analysis was conducted with the variables with a p-value of ≤0.2 in the univariable logistics regression analysis. We adjusted the multivariable logistic regression for education status, household income, marital status, physical activity, sedentary time, fruits and vegetable consumption, smokeless tobacco, self-reported diabetes and waist circumference for the male participants. We adjusted the multivariable logistic regression for place of residence, education status, household income, marital status, religion, physical activity, sedentary time, fruits and vegetable consumption, smokeless tobacco, and self-reported diabetes for the female participants. However, we did not include age, sex, smoking status, body mass index, and hypertension in the univariable or multivariable analyses as those were used as the parameters in the WHO CVD risk chart to calculate CVD risk. As we used those variables to calculate the same in our study, we did not use those variables in the regression analysis which provided us less option to add only place of residence, marital status and religion as the covariates in the analysis. 

Regarding stratification by sex, we mentioned earlier that we differentiated the entire analysis by sex considering the vulnerability, and poor health-care seeking practices of women in Bangladesh and similar settings to enable policymakers to focus on health-care seeking of both women and men while designing and implementing the healthcare program to address NCDs, especially cardiovascular diseases.

Discussion

- The sentence in lines 352-353 in difficult to understand and should be rephrased.

- Did the Authors consider to perform a lab-based analysis in a subgroup of their population? This could have validated their non-lab-based approach. Otherwise, the conclusions contained in lines 363-365 are too strong and should be rephrased.

Response: Noted with thanks. The objective of our paper was not to validate the non-laboratory-based CVD chart. Rather we estimated the CVD risk scores using the non-laboratory based chart as the World Health Organization has recommended the use of this chart in low-resource settings including Bangladesh and elsewhere where laboratory tests are not available. As the laboratory-based chart needs data on total cholesterol and the point or care test for total cholesterol is largely unavailable, the use of laboratory-based chart is still limited in the primary health care settings in Bangladesh. However, we have plan and are collecting data in another nationally representative study to compare two charts (lab-based and non-lab-based) in Bangladeshi population. 

- The paragraph between lines 389-396 is too general. Consider rephrasing.

- The intervention on CVD by primary physicians based on non-lab-based charts can only be hypothesized with this study, as follow up data were not collected. Therefore, the conclusions driven in lines 401-404 should be referred to as hypothetical.

References

- Ref 6,7, 15 and 19 are incomplete.

Response: Noted with thanks. We corrected the references. 

Reviewer #2: In the present paper, Abu Abdullah Mohammad Hanif and Colleagues aimed to investigate the role of a non-laboratory based cardiovascular disease (CVD) risk chart, recently developed by the World Health Organization (WHO)1, in assessing the prevalence of ten years CVD burdenamong Bangladeshi population aged 40-74 years.Parameters used in the risk algorithm were age, sex, smoking status, body mass index and systolic bloodpressure.They obtain six groups based on the calculated risk score:no risk (<5%), low risk (5% to <10%), moderate risk (10% to <20%), high risk (20% to <30%) and very high risk (≥30%).Then Authors listed potential factors associated with elevated CVD risk (defined as CVD risk ≥10%) among the subjects studied,based on literature review andconsidering data availability from this survey.Sociodemographic variables included place of residence (rural, non-slum urban, and slum), education, household income, marital status and religion; among the behavioralvariables were physical activity, sedentary time,fruits and vegetable consumption, and smokeless tobacco consumption status.So the study deals with an interestingissue concerning resource optimization topic; in low income countriesanon-laboratory based chart couldhelp the health care workers in screening, primary management counseling and referral of patients at higher risk of cardiovascular eventsto further evaluation. On the other hand, there are some methodological limitations that could weak the strength of these findings,in particular data accuracy and poor method description.

Major points

-In the method session about sample size and sampling techniques,Authors have selected patients according to origin area: rural, urban and slum zone. For each cluster research assistants listed households based on age groups; could author clarify age range and better defined it (year of age or age range as listed in table 1?). Moreover, it could be interesting screening patients also by gender in order to have balanced groups of the same age

Response: We added a flowchart describing steps of extracting the eligible participants for this analysis as the WHO CVD risk chart is applicable only for the persons aged 40-74 years. We also mentioned the age groups with the age in the method section. Please, see the Figure 1 and line numbers: 162-164.

-In the method session authors recorded “self-reported diabetes” if subjects had declared this comorbidity; it could be useful regarding the elevated risk of CVD in this cluster of patients also checking old blood chemistry exams, when available.

Response: We do not have data on blood chemistry. We are currently implementing another study as part of setting up a national NCD surveillance system and data from that project will enable us to explore more and compare laboratory-base and non-laboratory-based chart. 

-Method session need further explanation; authors refer to a non-laboratory based CVD risk chart, recently developed by the World Health Organization1; but in the aforementioned model were included participants without a known baseline history of cardiovascular disease; this aspect is not clarified in the study. Moreover,there are a lot of relevant parameters that are associated to an elevated risk of CVD, such as family history, chronic renal disease, carotid atherosclerosis,that have not been reported. All these variables are easily obtainable through an interview and have a significantly weight in the selection of clusters of subjects at higher risk of adverse events.

Response: We dropped the participants who had history of stroke during interview and reconducted the analysis. However, though we have data on history of heart diseases but there was not way of separating myocardial infarction or heart attck from other heart diseases. fRegarding other variables mentioned by the reviewer here, we didn’t collect data on those variables. We added relevant text to the limitations of this manuscript. Please see line 459-464. 

-In the “explanatory variables” session Authors defined outcome variable as elevated CVD risk if the total CVD risk score was ≥10% and CVD risk if the risk score was <10%; how they establish this cut-off?This choice needs further explanation.

Response: We followed the WHO categorization of CVD risks where CVD risk was considered as ‘very low or low risk’ if it is <10% and moderate to very high if the risk is >=10%. 

-In the Statistical analysis section Authors carry out bivariate logistic regression analysis, but they don’t itemize the variables sorted. Why did Authors employ a bivariable approach rather than a two-step univariable-multivariable logistic regression analysis?

Response: We indeed used univariable logistic regression to select variable for the multivariable logistic regression process. We corrected the texts in the revised manuscript. 

-In the result section Authors state that they extracted data of 7,757 males and females aged from 40 to 74 years old as the WHO CVD risk chart is only applicable for this age groups, please clarify this content, since in the referral model1 were included participants aged 40-80 years old. 

Response: Though the WHO CVD risk assessment chart working group used data from 40-80 years participants to derive, calibrate and validate the WHO CVD risk assessment charts, the WHO recommended chart is for the use of individuals between 40-74 years. Please, see the link below for accessing the charts: https://www.who.int/docs/default-source/ncds/cvd-risk-non-laboratory-based-charts.pdf?sfvrsn=fbb10584_2

-An explicative graphic of the study design could be helpful in evaluating patient screening from the initial 17,323 households to the effective 7,757 subjects studied.

Response: We added a flowchart demonstrating the process of extraction the eligible participants for analysis from entire study participants enrolled from 17,323 households. 

-In Table 1, are p-values referred to multivariable comparisons? Probably in this situation a sub-analysis for significantly different variables should be performed.

The p-values given in the Table-1 were NOT from multivariable comparisons, rather just showing the difference between the male female proportion of the participants for each of the background characteristics. However, to eliminate confusion, we dropped the p-values from Table-1 and revised the Table. 

-Could authors explain why in table 2 they considered only sociodemographic characteristics? -All results are itemized in four tables; this presentation method could make occult some relevant data; a figure about the prevalence of CVD in the different risk categories could focus on the topic of the paper.

Response: We added other variables in Table 2. We also added a figure [Figure 2] depicting elevated CVD risk by age and sex. 

-Could authors explain what are the multivariable analysis adjusted for and which is the covariate?

Response: Noted with thanks! We mentioned the variables we adjusted for in the multivariable logistic regression in the footnote under Table 3 and 4. The WHO non-laboratory based CVD risk assessment chart () uses age, sex, smoking status, BMI, and systolic blood pressure to estimate CVD risk in 10 years. As we used those variables to calculate CVD risk, we did not use those variables in the regression analysis. As a result, we hadplace of residence, martial status and religion as the covariates in the analysis. 

-In the discussion session Authors state “this analysis revealed that without even any laboratory procedures, the primary health care worker would be able to assess the CVD risk of the population and initiate management/referral procedure following national protocols for the CVD risk”(lines 363-365); without follow-up data and in the absence of a comparison with a lab-based analysis (also in a subgroup of their sample size) they could not sentencing this content.

Response: We revised the sentences mentioning the moderate agreement of the non-lab based chart with that of the lab-based chart. Please, check the lines: 404-409.

-In the discussion session the assertion: “insufficient physical activity may cause overweight, high blood pressure, and increase the level of bad cholesterol; tobacco can damage and narrow the blood vessels with harmful substances and, high blood sugar levels can damage the blood vessels, and all of these eventually cause CVDs”(lines 393-396) is too generic; please reward.

Response: We reworded the text. The text now reads: “Insufficient physical activity and other behavioral and clinical risk factors of NCDs, including overweight, high blood pressure, tobacco, and high blood sugar levels, are interrelated and can influence each other”

-Finally,considerations about the usefulness of a nonlab-based algorithm adopted by physicians for implementing subsequent prevention and management options (line 397-399) can only be hypothesized, since follow-up data are missing.

Response: we revised the sentence adding ‘supplement’ instead of ‘implement’. Please, see the line no. 443.

 Minor points

-Please correct line 294 where we read 189% vs. 16%

Response: Noted with thanks! We have corrected the number. Please, see the line number: 314.

-References are to be corrected

Response: Noted with thanks! We have corrected the references. 

-Please grammatically correct line 295 where probably verb is missing

Response: Noted with thanks! We have corrected it. 

-Please correct line 317: “,)”

Response: Noted with thanks! We have corrected it. 

-Please correct line 339: a bracket is missing

Response: Noted with thanks! We have corrected it. 

References

1) Kaptoge S, Pennells L, BacquerDD, Cooney MT, Kavousi M, Stevens G, et al. World Health Organization cardiovascular disease risk charts: revised models to estimate risk in 21 global regions. The Lancet Global Health. 2019;7(10):e1332-e45

---

## [Decision Letter · Decision Letter 1]

15 Apr 2021

PONE-D-20-10140R1

Ten-years cardiovascular risk among the Bangladeshi population using non-laboratory-based risk chart of the World Health Organization: findings from a nationally representative survey

PLOS ONE

Dear Dr. Mridha,

Thank you for submitting your manuscript to PLOS ONE. After careful consideration, we feel that it has merit but does not fully meet PLOS ONE’s publication criteria as it currently stands. Therefore, we invite you to submit a revised version of the manuscript that addresses all the minor points raised during the review process.

We look forward to receiving your revised manuscript.

Kind regards,

Giuseppe Vergaro, M.D.

Academic Editor

PLOS ONE

Journal Requirements:

Reviewers' comments:

Reviewer's Responses to Questions

**Comments to the Author**

1. If the authors have adequately addressed your comments raised in a previous round of review and you feel that this manuscript is now acceptable for publication, you may indicate that here to bypass the “Comments to the Author” section, enter your conflict of interest statement in the “Confidential to Editor” section, and submit your "Accept" recommendation.

Reviewer #1: All comments have been addressed

Reviewer #2: All comments have been addressed

2. Is the manuscript technically sound, and do the data support the conclusions?

Reviewer #1: Partly

Reviewer #2: Yes

3. Has the statistical analysis been performed appropriately and rigorously? 

Reviewer #1: Yes

Reviewer #2: Yes

4. Have the authors made all data underlying the findings in their manuscript fully available?

Reviewer #1: Yes

Reviewer #2: Yes

5. Is the manuscript presented in an intelligible fashion and written in standard English?

Reviewer #1: Yes

Reviewer #2: Yes

6. Review Comments to the Author

Reviewer #1: The Authors have reviewed extensively the paper and answered appropriately to the comments made by the Reviewers.

I have still two minor issues: the p values in the tables should be mentioned and not eliminated at all, now that the Tables have been clarified for the reader.

Also, in the Methods section, an effect size of 1.6 is mentioned, which is very wide. The sample size calculated might therefore be misleading. Please clarify.

Reviewer #2: The authors have adequately addressed my concerns. The study deals with an interesting issue regarding resource optimization topic. A non-laboratory based chart could help health care workers in screening, primary management counseling and referral of patients at higher risk cardiovascular events to further evaluation

7. PLOS authors have the option to publish the peer review history of their article (what does this mean?). If published, this will include your full peer review and any attached files.

Reviewer #1: No

Reviewer #2: No

---

## [Author Response · Author response to Decision Letter 1]

16 Apr 2021

PONE-D-20-10140R1

Ten-years cardiovascular risk among the Bangladeshi population using non-laboratory-based risk chart of the World Health Organization: findings from a nationally representative survey

We are thankful to the Editor and the reviewers for providing feedback on the revised manuscript. Below are the responses from the authors of this manuscript: 

Journal Requirements:

Response from author: We checked the reference list and corrected the incompleteness of some references we have found but as we have used used reference manager to correct the references, the changes do not reflect as tracked change. However, we did not include any new reference. We also checked each of the articles cited in our manuscript, and we did not find any retracted article. 

Reviewer #1: The Authors have reviewed extensively the paper and answered appropriately to the comments made by the Reviewers.

I have still two minor issues: the p values in the tables should be mentioned and not eliminated at all, now that the Tables have been clarified for the reader.

Response from author: Noted with thanks. We reinstated the p-value of the Chi square test conducted between sex and the listed characteristics of the study participants in table 1 and between risk categoris and the listed characteristics of the study participants in table 2. 

Also, in the Methods section, an effect size of 1.6 is mentioned, which is very wide. The sample size calculated might therefore be misleading. Please clarify.

Response from author: Noted with thanks. 

1.6 mentioned in the manuscript is not effect size, it is design effect. We are sorry for the confusion. Below is the clarification. The design effect (DEF), we calculated using the following formula: 

Design Effect = 1 + (n-1)*icc 

In our study, the cluster sample size was 62, and we the intra-cluster correlation was 0.01 [1]. 

Reviewer #2: The authors have adequately addressed my concerns. The study deals with an interesting issue regarding resource optimization topic. A non-laboratory based chart could help health care workers in screening, primary management counseling and referral of patients at higher risk cardiovascular events to further evaluation.

Response from author: Thank you so much!

References: 

1. Singh J, Liddy C, Hogg W, Taljaard M. Intracluster correlation coefficients for sample size calculations related to cardiovascular disease prevention and management in primary care practices. BMC Res Notes. 2015;8: 89. doi:10.1186/s13104-015-1042-y

---

## [Editor Report · Decision Letter 2]

7 May 2021

Ten-years cardiovascular risk among the Bangladeshi population using non-laboratory-based risk chart of the World Health Organization: findings from a nationally representative survey

PONE-D-20-10140R2

Dear Dr. Mridha,

We’re pleased to inform you that your manuscript has been judged scientifically suitable for publication and will be formally accepted for publication once it meets all outstanding technical requirements.

Kind regards,

Giuseppe Vergaro, M.D.

Academic Editor

PLOS ONE

---

## [Editor Report · Acceptance letter]

12 May 2021

PONE-D-20-10140R2 

Ten-years cardiovascular risk among the Bangladeshi population using non-laboratory-based risk chart of the World Health Organization: findings from a nationally representative survey  

Dear Dr. Mridha:

I'm pleased to inform you that your manuscript has been deemed suitable for publication in PLOS ONE. Congratulations! Your manuscript is now with our production department. 

Kind regards, 

on behalf of

Dr. Giuseppe Vergaro 

Academic Editor

PLOS ONE